# Protomer alignment modulates specificity of RNA substrate recognition by Ire1

**Weihan Li**[1,2†‡], **Kelly Crotty**[1,2†], **Diego Garrido Ruiz**[3], **Mark Voorhies**[4], **Carlos Rivera**[5], **Anita Sil**[4], **R Dyche Mullins**[2,6], **Matthew P Jacobson**[3], **Jirka Peschek**[1,2§*], **Peter Walter**[1,2*]

[1]Department of Biochemistry and Biophysics, University of California San Francisco, San Francisco, United States; [2]Howard Hughes Medical Institute, San Francisco, United States; [3]Department of Pharmaceutical Chemistry, University of California at San Francisco, San Francisco, United States; [4]Department of Microbiology and Immunology, University of California at San Francisco, San Francisco, United States; [5]Department of Molecular Biophysics and Biochemistry, Yale School of Medicine, New Haven, United States; [6]Department of Cellular and Molecular Pharmacology, University of California at San Francisco, San Francisco, United States

**\*For correspondence:**
jirka@walterlab.ucsf.edu (JP);
peter@walterlab.ucsf.edu (PW)

[†]These authors contributed equally to this work

**Present address:** [‡]Department of Anatomy and Structural Biology, Albert Einstein College of Medicine, Bronx, United States; [§] Biochemistry Center, Heidelberg University, Heidelberg, Germany

**Competing interests:** The authors declare that no competing interests exist.

**Abstract** The unfolded protein response (UPR) maintains protein folding homeostasis in the endoplasmic reticulum (ER). In metazoan cells, the Ire1 branch of the UPR initiates two functional outputs—non-conventional mRNA splicing and selective mRNA decay (RIDD). By contrast, Ire1 orthologs from *Saccharomyces cerevisiae* and *Schizosaccharomyces pombe* are specialized for only splicing or RIDD, respectively. Previously, we showed that the functional specialization lies in Ire1's RNase activity, which is either stringently splice-site specific or promiscuous (Li et al., 2018). Here, we developed an assay that reports on Ire1's RNase promiscuity. We found that conversion of two amino acids within the RNase domain of *S. cerevisiae* Ire1 to their *S. pombe* counterparts rendered it promiscuous. Using biochemical assays and computational modeling, we show that the mutations rewired a pair of salt bridges at Ire1 RNase domain's dimer interface, changing its protomer alignment. Thus, Ire1 protomer alignment affects its substrates specificity.

## Introduction

In eukaryotes, about one third of all proteins are folded in the endoplasmic reticulum (ER). The protein folding homeostasis of the ER is monitored and tightly regulated by a collective of signaling pathways, known as the unfolded protein response (UPR) (*Hetz et al., 2020*; *Walter and Ron, 2011*). The most evolutionarily conserved branch of the UPR is initiated by Ire1, an ER-transmembrane kinase/endoribonuclease (RNase). In response to accumulated unfolded proteins in the ER, Ire1 forms oligomers (*Aragón et al., 2009*; *Korennykh et al., 2009*; *Li et al., 2010*) and carries out two functional outputs. First, Ire1 initiates non-conventional splicing of *HAC1* (in *S. cerevisiae*) or *XBP1* (in metazoans) mRNA (*Cox et al., 1993*; *Mori et al., 1993*; *Sidrauski and Walter, 1997*; *Yoshida et al., 2001*). After cleavage by Ire1 and removal of the intron, the severed exons are ligated by tRNA ligase (*Jurkin et al., 2014*; *Kosmaczewski et al., 2014*; *Lu et al., 2014*; *Peschek et al., 2015*; *Peschek and Walter, 2019*; *Sidrauski et al., 1996*). The spliced mRNAs are translated into Hac1 and Xbp1 proteins, both of which are transcription factors that induce the UPR gene expression program in the nucleus (*Calfon et al., 2002*; *Travers et al., 2000*; *Van Dalfsen et al., 2018*; *Yoshida et al., 2001*). Second, Ire1 selectively cleaves a set of mRNAs that encode ER-targeted proteins. The cleaved mRNAs are subsequently degraded by the cellular RNA decay machinery (*Guydosh et al., 2017*). As a result, this process, known as regulated Ire1-dependent

mRNA decay (RIDD), restores homeostasis of the ER by reducing the protein folding burden (*Bae et al., 2019*; *Hollien et al., 2009*; *Hollien and Weissman, 2006*; *Kimmig et al., 2012*; *Moore and Hollien, 2015*).

While metazoan Ire1 performs both functions, the Ire1 orthologs in *Saccharomyces cerevisiae* and *Schizosaccharomyces pombe* are functionally specialized: *S. cerevisiae* Ire1 initiates splicing of *HAC1* mRNA as its singular target in the cell (*Niwa et al., 2005*), and *S. pombe* Ire1 exclusively performs RIDD (*Guydosh et al., 2017*; *Kimmig et al., 2012*). Our previous study reported that the functional specialization of Ire1 is achieved through diverged RNase specificities (*Li et al., 2018*). *S. cerevisiae* Ire1 has a stringent RNase, restricting it to *HAC1* mRNA. In contrast, *S. pombe* Ire1 has a promiscuous (i.e. broadly specific) RNase, enabling cleavage of a wide range of mRNA RIDD targets. Which structural determinants on Ire1 influence RNase specificity remained unknown. Here, we addressed this question by mutagenesis-guided biochemical analyses and structural modeling.

## Results

### RNase activity of *S. pombe* Ire1 is toxic to bacterial cells

We recently purified and characterized recombinant *S. cerevisiae (Sc)* and *S. pombe (Sp)* Ire1 kinase/RNase (KR) domains (*Li et al., 2018*). During the protein expression process, we noticed that the presence of plasmids bearing the genes encoding *Sc* and *Sp* Ire1-KR under the control of the T7 promoter differently affected growth of the *E. coli* host cells. Growth curves revealed that *E. coli* cells bearing a plasmid containing the *Sp IRE1-KR* barely grew within the monitored 5 hr time window, even in the absence of the isopropyl β-d-1-thiogalactopyranoside (IPTG) inducer (*Figure 1A*, blue filled triangles). By contrast, *E. coli* cells bearing plasmids containing *Sc IRE1*-KR grew normally with a growth rate comparable to that of a control strain bearing an empty plasmid. Because the T7 promoter is known to exhibit background expression even in the absence of IPTG (*Rosano and Ceccarelli, 2014*), we reasoned that the observed gene toxicity of *Sp IRE1-KR* might result from the enzyme's promiscuous RNase activity, which may degrade endogenous *E. coli* RNAs required for viability. By contrast, *Sc* Ire1-KR might be tolerated, because of its exquisite substrate specificity for *Sc HAC1* mRNA splice junctions (*Niwa et al., 2005*). To test this notion, we used the Ire1 RNase inhibitor 4μ8C (*Cross et al., 2012*). As expected, 4μ8C inhibited cleavage of a 21 base-pair stem-loop substrate derived from the 3' splice junction of *XBP1* mRNA for both *Sp* and *Sc* Ire1-KR (*Figure 1B and C*, *Figure 1—figure supplement 1A–C*), as well as RIDD activity in *Sp* cells (*Figure 1—figure supplement 1D*). Importantly, when added to the cultures of cells bearing plasmids encoding *Sp* Ire1-KR, 4μ8C restored normal growth (*Figure 1A*). In further agreement with the notion that *Sp* RNase activity was the culprit of reduced *E. coli* growth, an RNase-dead mutant of *Sp* Ire1-KR harboring the H1018A mutation (*Kimmig et al., 2012*) was not toxic (*Figure 1A*). These results suggest that *E. coli* growth was inhibited by the endonuclease activity, rather than by indirect effects, such as protein misfolding or aggregation. Moreover, we established that bacterial growth can be exploited to assess substrate specificity of Ire1's RNase.

### Ire1's RNase domain confers promiscuous RNase activity

We reasoned that this assay might allow us to glean insights into Ire1's substrate specificity. Using structure-guided sequence comparison of Ire1 RNase domains, we picked a total of seventeen residues whose common features include that they are (i) part of an oligomerization interface or located within 18 Å from the helix-loop element (HLE), which contains a positively charged loop (N1036 to K1042 on *Sc* Ire1) that engages the RNA substrates (*Korennykh et al., 2009*; *Korennykh et al., 2011*; *Lee et al., 2008*), and (ii) divergent amino acids between *Sc* and *Sp* but conserved within the Schizosaccharomyces genus. Among the seventeen amino acids, seven are located near the HLE, five are located at the RNase-RNase interface within the back-to-back dimer (previously defined as interface IF1[C] *Korennykh et al., 2009*), and five are located at the RNase-RNase interface in the active Ire1 oligomer (previously defined as interface IF2[C] *Korennykh et al., 2009*; *Figure 2A & B*, *Figure 2—source data 1*).

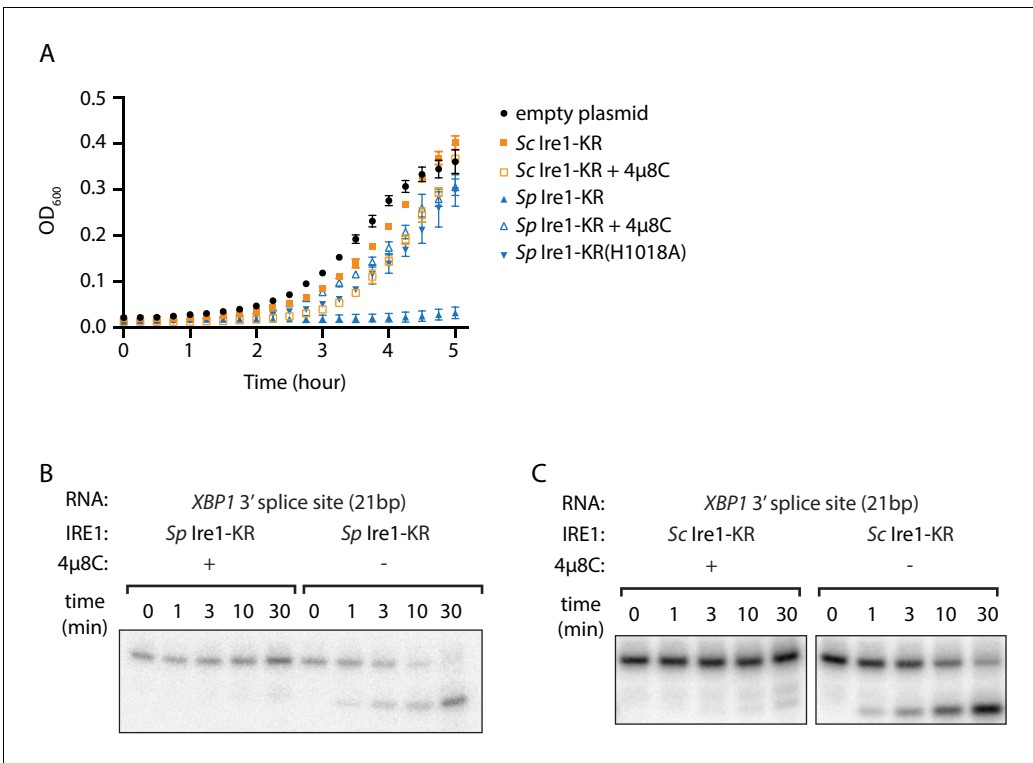

**Figure 1.** The promiscuous RNase activity of *S. pombe* Ire1 causes toxicity to bacterial cells. (**A**) Growth curves of bacterial cells expressing various Ire1 kinase-RNase (KR) domains. Optical densities at 600 nm ($OD_{600}$) were measured every 15 min for 5 hr. Bacterial cells expressing *S. cerevisiae* (*Sc*) or *S. pombe* (*Sp*) Ire1-KR were incubated at 37°C. In the indicated samples, 1 μM of the Ire1's RNase inhibitor 4μ8C was added to the culture. The *Sp* Ire1-KR(H1018A) has a catalytically inactive RNase. (**B, C**) In vitro RNA cleavage assays with or without 200 μM of 4μ8C. 5′ radiolabeled stem-loop RNA substrates, which are derived from the *XBP1* mRNA 3′ splice site, were incubated with 12.5 μM of *Sp* (**B**) or *Sc* (**C**) Ire1-KR at 30°C for the indicated time.

The online version of this article includes the following figure supplement(s) for figure 1:

**Figure supplement 1.** 4μ8C inhibits the RNase activity of *S. pombe* and *S. cerevisiae* Ire1.

To examine possible effects of these residues on RNase specificity, we cloned and purified an *Sc* Ire1-KR mutant with all 17 residues replaced by their *S. pombe* counterparts (*Sc* Ire1-KR-mut17). We tested its RNase activity using four previously characterized stem-loop RNA substrates derived from the *Sc HAC1* mRNA 3′ splice site that is exclusively cleaved by *Sc* Ire1, as well as the Ire1 cleavage sites of the *Sp BIP1*, *PLB1* and *SPAC4G9.15* mRNAs that are exclusively cleaved by *Sp* Ire1 (*Li et al., 2018*). We chose the three *Sp* RNA substrates because their Ire1 cleavage sites vary in predicted loop sizes (9-, 7-, and 3-membered loops, respectively). Cleavage activity towards stem-loop RNAs with variable loop sizes is one of the characteristic features of Ire1 promiscuity (*Li et al., 2018*). In agreement with previous results, wildtype (WT) *Sc* Ire1-KR cleaved the *HAC1* mRNA 3′ splice site but none of the *Sp* stem-loop RNA substrates. Remarkably, *Sc* Ire1-KR-mut17 efficiently and specifically cleaved all four stem-loop RNA substrates (*Figure 2C–F*) with comparable kinetics (*Figure 2G* and *Figure 2—figure supplement 1*). These results show that introducing the 17 mutations made *Sc* Ire1 more 'pombe-like' regarding its acceptance of variable-loop RIDD substrates. When tested in the *E. coli* growth assay, the expression of Ire1-KR-mut17 proved toxic and 4μ8C alleviated the toxicity (*Figure 2H*), confirming the notion that toxicity results from the enzyme's broadened substrate range.

## Residues at Ire1's dimer interface confer RNase promiscuity

The above results demonstrate that bacterial growth can be a useful readout for Ire1's RNase promiscuity. Hence, we used this assay to identify the residue(s) that were causal in conferring the

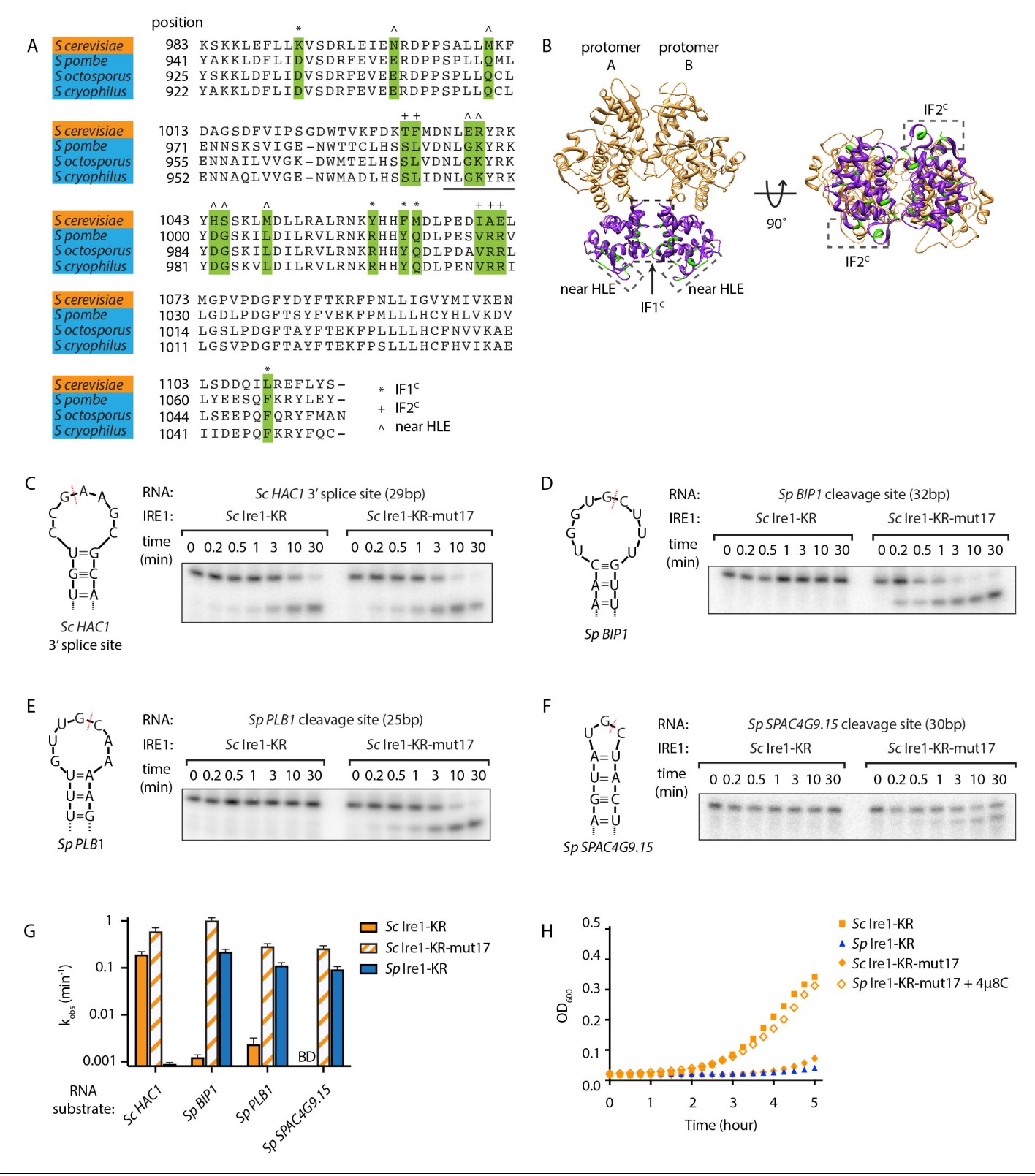

**Figure 2.** *S. cerevisiae* Ire1-KR-mut17 has a promiscuous RNase activity. (**A**) Sequence alignment of the RNase domains of Ire1 orthologs from *Saccharomyces cerevisiae*, *Schizosaccharomyces pombe*, *Schizosaccharomyces octosporus*, *Schizosaccharomyces cryophilus*. A total of 17 residues (green) were selected as the candidate residues that may regulate Ire1's RNase promiscuity. These candidate residues are located at back-to-back dimer interface (marked with *), oligomer interface (marked with +) or near the helix loop element (marked with ^). Sequence of the helix loop element

*Figure 2 continued on next page*

*Figure 2 continued*

(HLE) is underlined. (**B**) The location of the 17 candidate residues (green) on the back-to-back dimer structure of the Ire1 cytosolic domain (PDB: 3FBV) with kinase domain in yellow and RNase domain in purple. The dimer interface, oligomer interface and HLE regions are indicated in dashed boxes. (**C–F**) In vitro RNA cleavage assays with 12.5 µM of wildtype (WT) *Sc* Ire1-KR or *Sc* Ire1-KR-mut17. The stem-loop RNA substrates are derived from the *Sc HAC1* mRNA 3' splice site (**C**), *Sp BIP1* (**D**), *PLB1* (**E**), and *SPAC4G9.15* (**F**) mRNA cleavage sites. Experimental conditions are the same as in *Figure 1B&C*. Predicted RNA secondary structures are illustrated. Ire1 cleavage sites are marked with red dashed lines. (**G**) Comparison of the $k_{obs}$ of WT *Sc* Ire1-KR, *Sc* Ire1-KR-mut17 and *Sp* Ire1-KR. The $k_{obs}$ of WT *Sc* Ire1-KR and *Sc* Ire1-KR-mut17 were calculated from experiments in (**C–F**). The $k_{obs}$ of *Sp* Ire1-KR was measured under the same condition from our previous study (*Li et al., 2018*). 'BD' indicates cleavage activity below detection limit. Experiments were performed in duplicates. (**H**) Bacterial growth assay for WT *Sc* Ire1-KR, *Sc* Ire1-KR-mut17, and *Sp* Ire1-KR. Experimental conditions are the same as in *Figure 1A*. In the indicated samples, 1 µM of 4µ8C was added.

The online version of this article includes the following source data and figure supplement(s) for figure 2:

**Source data 1.** A list of the 17 candidate residues on *S. cerevisiae* and *S. pombe* Ire1.

**Figure supplement 1.** Quantification of in vitro cleavage assays.

broadened substrate specificity of *Sc* Ire1-KR-mut17. To this end, we created single revertants, each with one of the seventeen mutations converted back to the original amino acid in *Sc* (*Figure 3— source data 1*). We expected that when we revert a mutation that contributes to the enzyme's broadened substrate specificity, the revertant would be stringent and less toxic to *E. coli* cells. We found that three revertants (K992, H1044, and Y1059) showed markedly reduced toxicity, whereas the other 14 revertants remained toxic (*Figure 3A*). Next, we cloned and purified a mutant protein, *Sc* Ire1-KR(K992D,H1044D,Y1059R), in which we combined the three identified mutations. *Sc* Ire1-KR (K992D,H1044D,Y1059R) cleaved both *Sp* stem-loop and *Sc HAC1* 3' splice-site substrates efficiently (*Figure 3B–E*) and with similar rates (*Figure 3F* and *Figure 3—figure supplement 1*). These results narrowed the list of candidate amino acid changes that confer RNase promiscuity down to three.

Based on the locations of these three residues, we divided them into two groups. *Sc* Ire1 K992 and Y1059 (corresponding to D950 and R1016 of *Sp* Ire1) are located at Ire1's back-to-back dimer interface, while *Sc* Ire1 H1044 (corresponding to *Sp* Ire1 D1001) is located two amino acids C-terminal of the HLE. We cloned and purified two Ire1 mutants, *Sc* Ire1-KR(K992D,Y1059R) and *Sc* Ire1-KR (H1044D). Recombinantly expressed and purified *Sc* Ire1-KR(K992D,Y1059R) cleaved both *Sc* and *Sp* stem-loop RNA substrates with efficiencies comparable to those of *Sc* Ire1-KR(K992D,H1044D, Y1059R) (*Figure 3B–F*). By contrast, *Sc* Ire1-KR(H1044D) cleaved the cognate *Sc* RNA substrate but none of the *Sp* RNA substrates (*Figure 3B–F*), suggesting that the two *cerevisiae*-to-*pombe* mutations at Ire1's RNase-RNase dimer interface confer Ire1 RNase promiscuity. By contrast, H1044D appears a false positive, likely isolated because the bacterial assay cannot distinguish stringent RNase from inactive RNase as neither is toxic to the bacterial cells.

## *S. cerevisiae* Ire1-KR(K992D,Y1059R) recognizes RNA substrates with reduced stringency

*Sc* Ire1 displays a strong preference for RNA substrates that contain a consensus sequence within a stem-loop structure (*Gonzalez et al., 1999*; *Hooks and Griffiths-Jones, 2011*; *Li et al., 2018*; *Oikawa et al., 2010*). We next characterized the RNA motif recognized by *Sc* Ire1-KR(K992D, Y1059R) and compared it to those recognized by WT *Sc* and *Sp* Ire1-KR. To this end, we examined Ire1 cleavage efficiencies on a series of *HAC1*- and *BIP1*-derived mutant stem-loop RNAs, in which each loop residue was individually changed into the three other possible ribonucleotides (*Figure 4A & B*). Using the *HAC1*-derived mutant substrates, we showed that WT *Sc* Ire1-KR showed specificity for the sequence motif CNG|(C/A)NGN, in close agreement with previous findings (*Gonzalez et al., 1999*). By comparison, *Sc* Ire1-KR(K992D,Y1059R) recognized a less-stringent sequence motif, CNG| NNGN, in particular tolerating base substitutions in the +1 position (*Figure 4A*).

To assess the effects of RNA loop size variation on Ire1 cleavage efficiency, we engineered mutations at positions −4 to break the base-pairing at the tip of the stem and enlarge the loop from 7 to 9 nucleotides. The 9-membered stem loops were not cleaved by WT *Sc* Ire1-KR, in line with previous study (*Gonzalez et al., 1999*). By contrast, the same RNAs were cleaved by *Sc* Ire1-KR(K992D, Y1059R) (*Figure 4A*). The results suggest that the two interface mutations in *Sc* Ire1-KR(K992D, Y1059R) render the enzyme more tolerant to both RNA sequence and loop size variations.

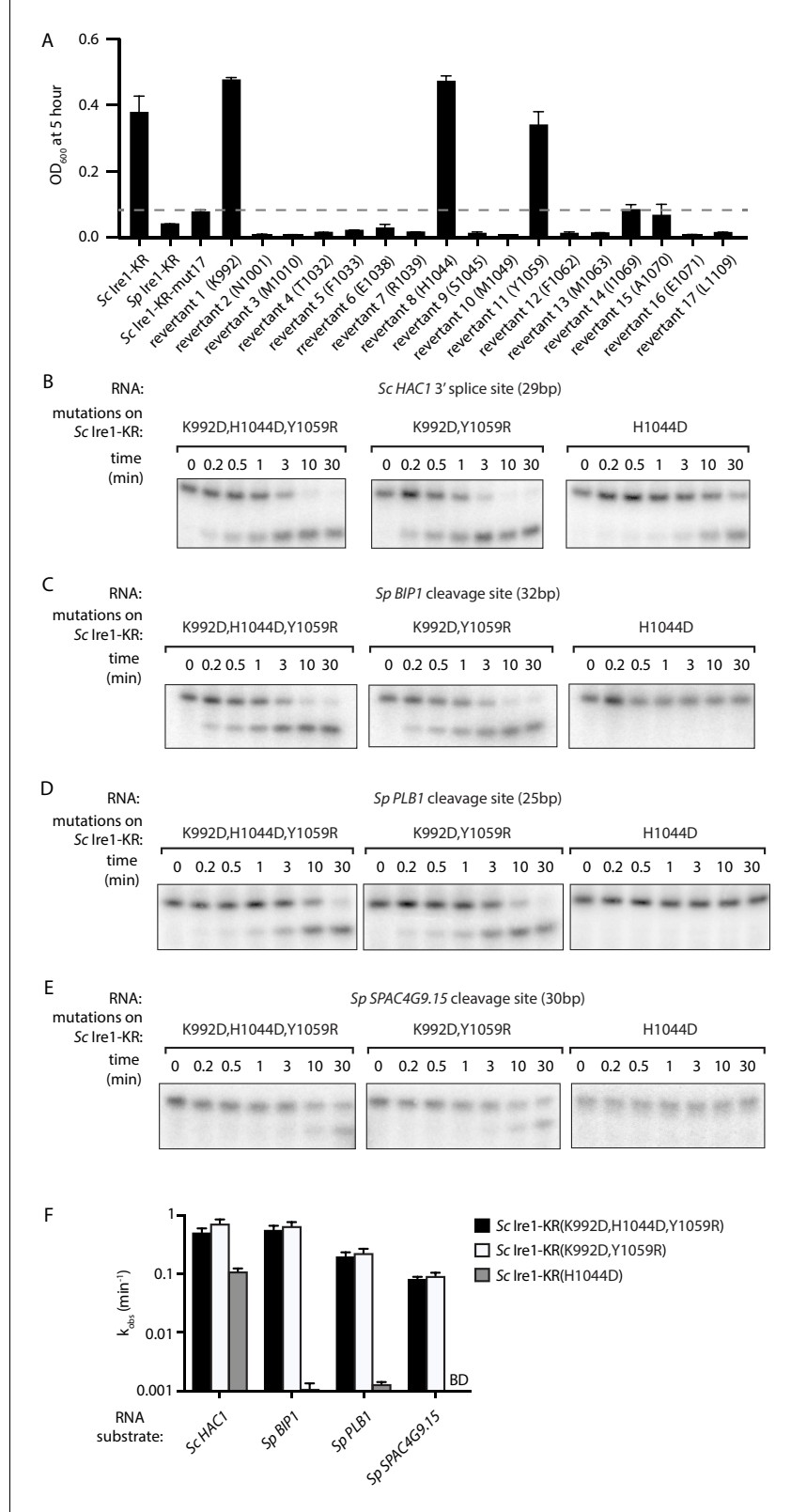

**Figure 3.** Two residues at Ire1's RNase-RNase dimer interface regulate Ire1's RNase promiscuity. (**A**) Bacterial growth assay for *Sc* Ire1-KR revertants. Conditions are the same as in *Figure 1A*. OD$_{600}$ at 5 hr time-point was measured. Experiments were performed in duplicates. Dashed line marks the threshold used to separate toxic and non-toxic Ire1 constructs. (**B–E**) In vitro cleavage assays of *Sc* Ire1-KR(K992D,H1044D,Y1059R), *Sc* Ire1-KR(K992D, *Figure 3 continued on next page*

*Figure 3 continued*

Y1059R) and *Sc* Ire1-KR(H1044D) on *Sc HAC1* mRNA 3' splice site (B), *Sp BIP1* (C), *PLB1* (D) and *SPAC4G9.15* (E) mRNA cleavage sites. Experimental conditions are the same as in *Figure 1C*. (F) Comparison of the $k_{obs}$ that are calculated from results in (B–E). 'BD' indicates cleavage activity below detection limit. Experiments were performed in duplicates.

The online version of this article includes the following source data and figure supplement(s) for figure 3:

**Source data 1.** The detailed sequence information of *S. cerevisiae* Ire1-KR constructs used in this study.

**Figure supplement 1.** Quantification of in vitro cleavage assays.

The Ire1 cleavage site in *Sp BIP1* mRNA contains a 9-nucleotide loop with a UG|C cleavage site shifted by one nucleotide (*Figure 4B*). We confirmed that a stem-loop RNA substrate containing this site was efficiently cleaved by *Sp* Ire1-KR but not by *Sc* Ire1-KR, in agreement with our previous reports (*Guydosh et al., 2017*; *Kimmig et al., 2012*; *Li et al., 2018*). By contrast, *Sc* Ire1-KR(K992D, Y1059R) cleaved the motif efficiently with the additional tolerance of any nucleotide at position −2, thus reducing the required sequence motif to only G and C nucleotides flanking the cleavage site (*Figure 4B*). G|C is likewise present at the *PLB1* and *SPAC4G9.15* mRNA cleavage sites embedded in 7- and 3-membered loops, respectively, which also proved to be substrates of *Sc* Ire1-KR(K992D, Y1059R) (*Figure 3B–E*). In further support of the notion that *Sp* Ire1-KR and *Sc* Ire1-KR(K992D, Y1059R) are tolerant to loop size variation, two of our stem-loop RNA substrates harboring mutations, $U_{-5} \rightarrow A$ and $U_{+4} \rightarrow A$, respectively, which are predicted to contract the 9-membered loop to a 7-membered one, were efficiently cleaved by both enzymes (*Figure 4B*). Together, these data affirm the notion that *Sp* Ire1-KR and *Sc* Ire1-KR(K992D,Y1059R) are promiscuous enzymes that recognize short RNA sequence motifs and can accept a range of loop sizes.

## Salt-bridge rewiring at Ire1's dimer interface changes Ire1's protomer alignment

To understand how the interface mutations confer promiscuous RNase activity, we explored structural differences between *Sp* Ire1, *Sc* Ire1, and *Sc* Ire1-KR(K992D,Y1059R) using molecular modeling. Active *Sc* Ire1 oligomers are composed of multiple Ire1 back-to-back dimers that stack in a helical arrangement (*Korennykh et al., 2009*). K992 and Y1059 are located at the RNase-RNase interface of the back-to-back assembly of Ire1 protomers in PDB 3FBV (protomer A and B in *Figure 5A*). To build a structural model of *Sc* Ire1-KR(K992D,Y1059R), we introduced K992D and Y1059R onto the *Sc* Ire1 dimer structure and performed energy minimization to optimize distances and resolve steric clashes. We followed this calculation with molecular dynamics (MD) simulations, comparing *Sc* Ire1-KR(K992D,Y1059R) and WT *Sc* Ire1-KR. Analysis was performed from 10 ns to 20 ns, and the simulation structures reached equilibrium within 10 ns (*Figure 5—figure supplement 1*). The convergent structure model of *Sc* Ire1-KR(K992D,Y1059R) predicts a structural rearrangement at Ire1's dimer interface. Specifically, whereas in the WT *Sc* Ire1-KR dimer residues E988 and K992 of both protomers form a symmetric pair of salt bridges across the dimer interface (*Figure 5B*), these salt bridges are absent in the dimer of *Sc* Ire1-KR(K992D,Y1059R) due to the charge reversal introduced by the K992D mutation. More interestingly, the model predicts the formation of a new pair of salt bridges in *Sc* Ire1-KR(K992D,Y1059R), connecting the two newly introduced amino acids D992 and R1059 across the protomer/protomer interface (*Figure 5C*). The MD simulation predicts these new bonds as stable features (*Figure 5—figure supplement 2*). Thus, molecular modeling suggests a structural rearrangement, resulting from the two interface mutations in *Sc* Ire1, which, we propose, allows *Sc* Ire1 to assume the promiscuous '*pombe*-like' state (*Figure 5D*).

The predicted new salt bridges in *Sc* Ire1-KR(K992D,Y1059R) are mediated by guanidinium-carboxylate bidentate interactions, which are among the strongest non-covalent interactions in proteins and are considerably stronger than the ammonium-carboxylate interaction seen in the salt bridges in WT *Sc* Ire1-KR (*Masunov and Lazaridis, 2003*). Given that the interaction at the dimer interface is predicted to be stronger in mutant *Sc* Ire1-KR(K992D,Y1059R) than in WT *Sc* Ire1-KR, the mutant enzyme should be more prone to form dimers/oligomers than WT *Sc* Ire1-KR. Indeed, we confirmed this notion using protein crosslinking in vitro, followed by SDS-PAGE, showing that WT *Sc* Ire1-KR was mostly monomeric, while *Sc* Ire1-KR(K992D,Y1059R) formed mostly dimers and tetramers

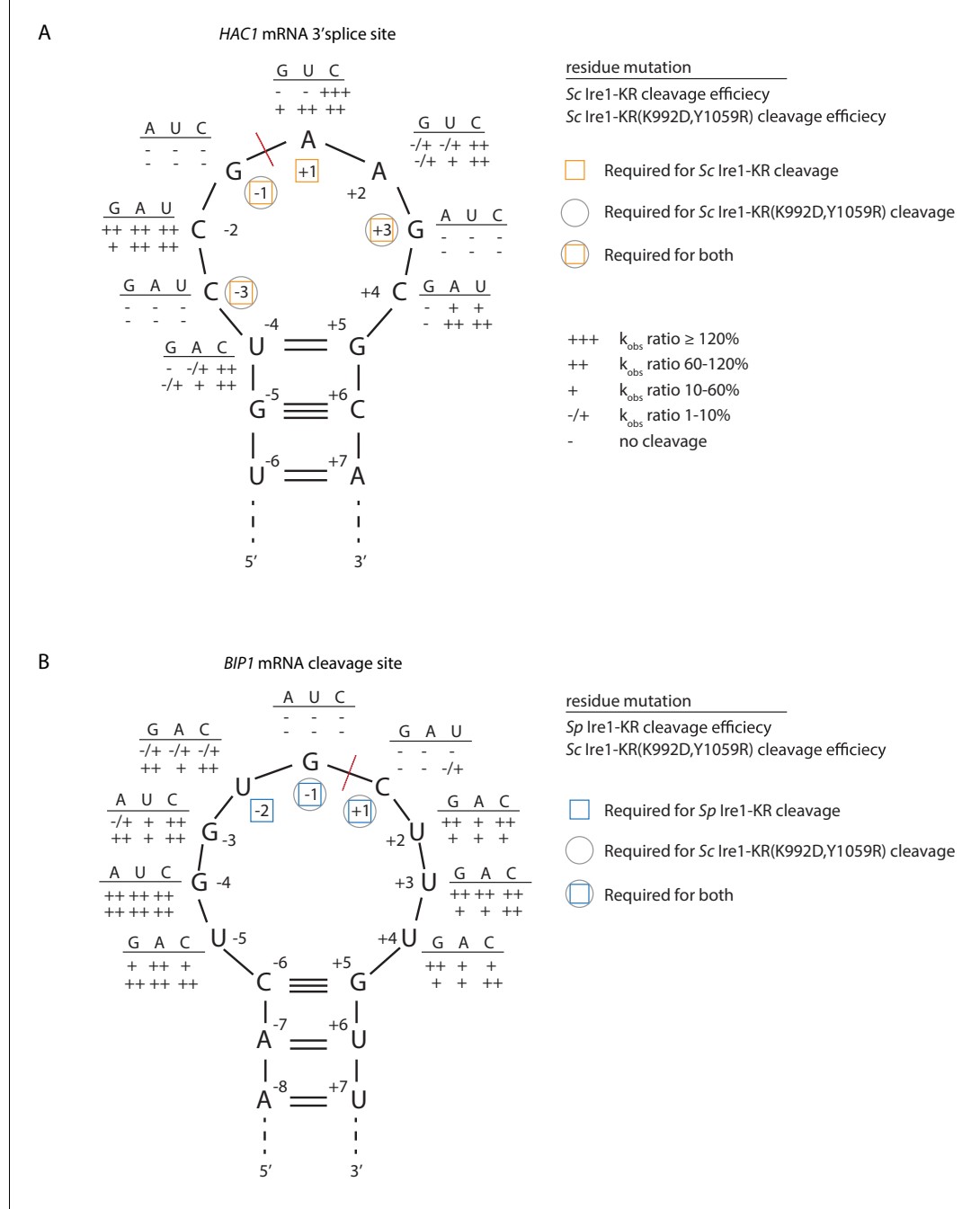

**Figure 4.** *S. cerevisiae* Ire1-KR(K992D,Y1059R) has a promiscuous RNase activity. (**A–B**) A series of twenty-four (**A**) and twenty-seven (**B**) stem-loop RNA substrates, which are derived from the *S. cerevisiae HAC1* mRNA 3' splice site (**A**) or the *S. pombe BIP1* mRNA cleavage site (**B**), are in vitro transcribed. Each of the substrate carries a single-point mutation, which is located on the loop or at the end position of the stem. The sequence of the various point mutations is indicated next to each residue (above the line). Listed below these sequences are the cleavage efficiencies, at which each mutant RNA substrate was cleaved by *Sc* Ire1-KR (first row below the line in panel A), *Sp* Ire1-KR (first row below the line in panel B) or *Sc* Ire1-KR(K992D,Y1059R) (second rows below the lines in panels A and B). $k_{obs}$ of mutant stem-loop RNAs is normalized to $k_{obs}$ of WT *HAC1* (**A**) or *BIP1* (**B**) stem-loop RNAs that are cleaved by the corresponding Ire1. (-) is no cleavage; (-/+) is 1–10%; (+) is 10–60%; (++) is 60–120%; (+++) is >120%. Ire1 cleavage sites are marked with red dashed lines. Yellow squares (in A), blue squares (in B) and gray circles (in A and B) mark the positions, at which specific residues are required to achieve efficient cleavages by *Sc* Ire1-KR, *Sp* Ire1-KR, and *Sc* Ire1-KR(K992D,Y1059R), respectively. Experimental conditions are the same as in *Figure 1C*.

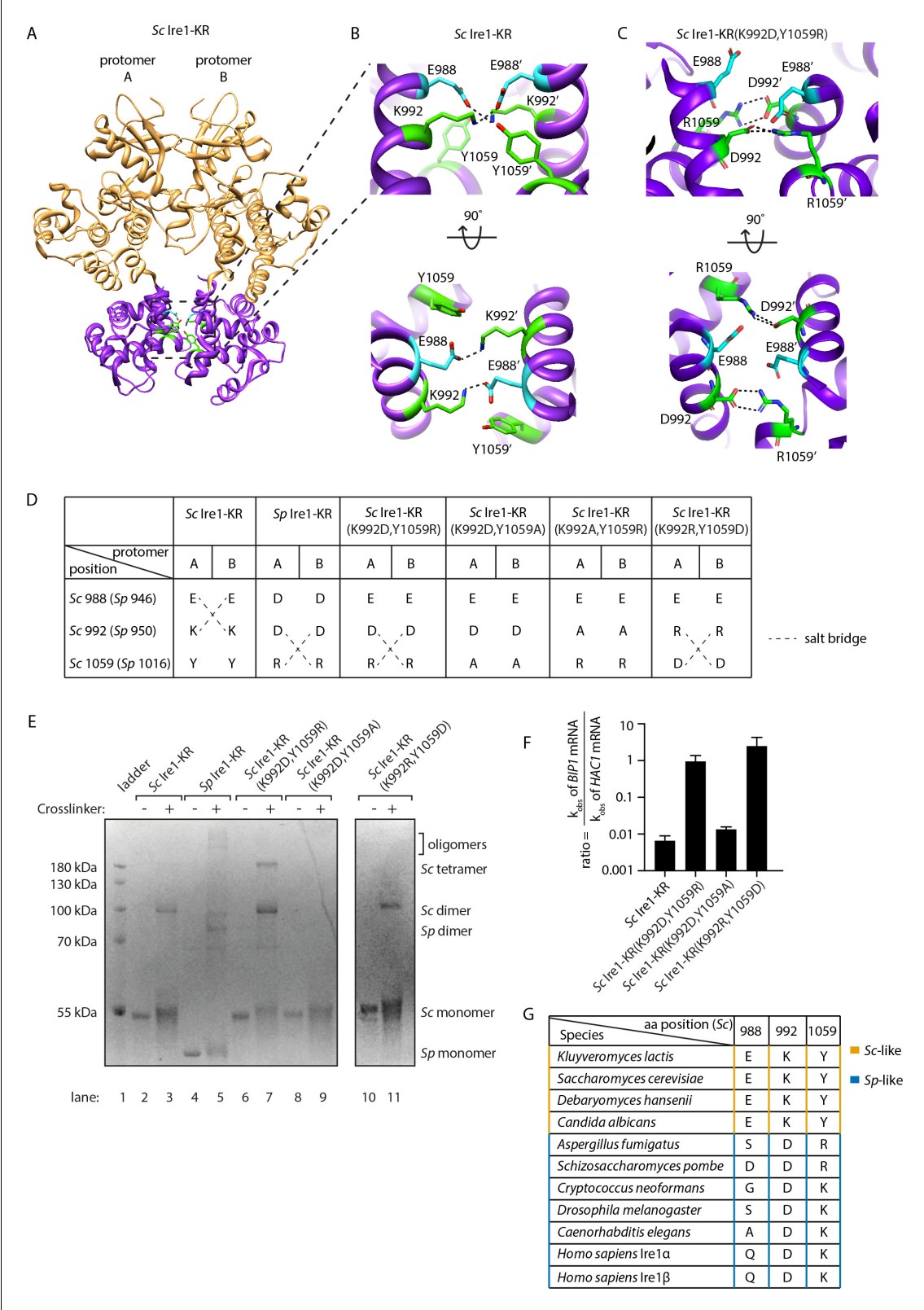

**Figure 5.** Structural re-arrangement at Ire1 dimer interface regulates the RNase promiscuity. (**A**) Back-to-back dimer structure of WT *Sc* Ire1 cytosolic domain (PDB: 3FBV) with kinase domain in yellow and RNase domain in purple. K992 and Y1059 are colored in green while E988 is colored in blue. Side chain labels on protomer B are marked with '. (**B**) Close-up view focusing on the interface region of WT *Sc* Ire1 dimer. Dashed lines indicate salt bridges. (**C**) Close-up view focusing on the interface region of the predicted dimer structure of *Sc* Ire1-KR(K992D,Y1059R), which was generated by a 20-

*Figure 5 continued on next page*

*Figure 5 continued*

ns molecular dynamics (MD) simulation from an initial structural model that was built from the WT *Sc* Ire1 dimer (PDB: 3FBV). The final frame of the simulation was illustrated. D992 and R1059 are colored in green while E988 is colored in blue. Dashed lines indicate salt bridges. (D) Illustration of the residues at *Sc* Ire1 position 988, 992, and 1059 (or *Sp* Ire1 position 946, 950 and 1016). Dashed lines indicate salt bridges. (E) Crosslinking gel for various Ire1-KR constructs. Indicated Ire1-KR (12.5 μM) constructs were incubated with or without 1 mM of crosslinker bissulfosuccinimidyl suberate for 2 hr on ice before being separated on an SDS-PAGE gel and stained by coomassie blue. (F) Ire1's ability to distinguish *Sc HAC1*- and *Sp BIP1*-derived RNA substrates is measured by the ratio of their corresponding $k_{obs}$. (G) Evolutionary comparison of Ire1 orthologs from various species. The analysis focuses on three residues, which correspond to position 988, 992 and 1059 on *Sc* Ire1. The *S. cerevisiae*-like pattern (yellow) has an E/D at 988, K/R at 992 and Y at 1059. The *S. pombe*-like pattern (blue) has a varying amino acid (aa) at 988, D/E at 992 and K/R at 1059.

The online version of this article includes the following source data and figure supplement(s) for figure 5:

**Source data 1.** In this table, 230 Ire1 orthologs were compared.
**Figure supplement 1.** The root-mean-square deviation (RMSD) of atomic positions of Ire1 RNase domain during the simulation.
**Figure supplement 2.** Time fraction of the MD simulation during which the indicated salt bridges are present.
**Figure supplement 3.** Sedimentation equilibrium analytical ultracentrifugation (SE-AUC) analysis of *Sc* Ire1-KR and *Sc* Ire1-KR(K992D,Y1059R).
**Figure supplement 4.** RNA cleavage efficiencies of *Sc* Ire1-KR mutants bearing mutations at back-to-back dimer interface.
**Figure supplement 5.** Evolutionary comparison of Ire1 orthologs.

(*Figure 5E*, compare lanes 2 and 3 with lanes 6 and 7). We further verified this result using sedimentation equilibrium analytical ultracentrifugation (*Figure 5—figure supplement 3*). For *Sc* Ire1-KR (K992D,Y1059R), we calculated a dissociation constant ($K_D$) of 0.98 μM, which is about 60-fold smaller than that of WT *Sc* Ire1-KR ($K_D$ = 57 μM). Based on these results, we consider it likely that the estimated gain in free energy of the predicted new salt bridges results in higher affinity within the back-to-back dimer, thus increasing the propensity of Ire1 to oligomerize.

Next, we experimentally tested the predicted salt bridges using mutagenesis. To this end, we first engineered *Sc* Ire1-KR(K992D,Y1059A), introducing an alanine at position 1059. This Ire1 mutant can neither form *S. cerevisiae*-like nor *S. pombe*-like salt bridges (*Figure 5D*). Thus, as expected, *Sc* Ire1-KR(K992D,Y1059A) did not form dimers (*Figure 5E*, lanes 8 and 9) and displayed ~100 fold reduced cleavage efficiency on *HAC1*-derived and an additional 100-fold (i.e., overall ~10,000 fold) reduced cleavage rate on *BIP1*-derived RNA substrates (*Figure 5—figure supplement 4*). Thus, surprisingly, breaking the pombe-like salt bridge arrangement restored *Sc* Ire1's ability to discriminate between substrate RNAs by ~100 fold (*Figure 5F*).

Breaking the predicted salt bridges on *Sc* Ire1-KR(K992D,Y1059R) by mutating aspartate 992 to alanine also abolished Ire1's RNase activity (*Sc* Ire1-KR(K992A,Y1059R); *Figure 5—figure supplement 4*), in this case reducing activity towards both *HAC1*- and *BIP1*-derived substrate RNAs beyond our detection limit (>10,000 fold). Thus, we were not able to assess substrate specificity for this mutant.

Finally, we generated a charge-reversal mutant of *Sc* Ire1-KR(K992D,Y1059R) by changing aspartate 992 to arginine and arginine 1059 to aspartate *Sc* Ire1-KR(K992R,Y1059D). We expected these two mutations to restore the salt bridges predicted for *Sc* Ire1-KR(K992D,Y1059R) but with reversed polarity. We found that, while the overall cleavage rate remained 100-fold suppressed for both RNA substrates, *Sc* Ire1-KR(K992R,Y1059D) regained activity towards the *BIP1*-derived substrate (*Figure 5F*, *Figure 5—figure supplement 4*) and formed dimers (*Figure 5E*, lanes 10 and 11). Together, these experiments validate the predicted salt bridges on *Sc* Ire1-KR(K992D,Y1059R) and further underscore the notion that salt bridge rewiring to a pombe-like arrangement confers promiscuity to Ire1's RNase activity.

The importance of both types of salt bridges is further highlighted by a sequence comparison of 230 Ire1 orthologs from yeast to human (see *Figure 5G* for a partial list of the Ire1 orthologs; a complete list is included in *Figure 5—figure supplement 5* and *Figure 5—source data 1*). We compared residues at three positions corresponding to the *Sc* Ire1 E988, K992 and Y1059. We found that 175 out of 230 of the Ire1 orthologs, including human Ire1α and Ire1β, have the *S. pombe*-like pattern, characterized by significant amino acid variation at position 988, a negatively charged amino acid (aspartate or glutamate) at position 992, and a positively charged amino acid (lysine or arginine) at position 1059. The apparent co-evolution of position 992 and 1059 further supports the existence of an inter-molecular salt bridge. Thirty-three out of 230 of the Ire1 orthologs have the *S. cerevisiae*-like pattern—with a negatively charged amino acid (aspartate or glutamate) at position 988, a

positively charged amino acid (lysine or arginine) at position 992 and, in most cases, a tyrosine at position 1059. Co-evolution of position 988 and 992 supports their interaction at the dimer interface.

## Interface mutations change the protomer alignment in Ire1 dimer

To gain an appreciation of how the interface mutations affect Ire1's active RNase site in the dimer, we compared the structure of WT *Sc* Ire1-KR dimer with the predicted structure of *Sc* Ire1-KR (K992D,Y1059R) dimer after aligning the two KR dimers by one protomer (*Figure 6A and B* for front and bottom-up view, respectively). Interestingly, we observed a rocking motion between the RNase domains of the protomers (*Figure 6B*). Specifically, the salt bridge between D992 and R1059' in *Sc* Ire1-KR(K992D,Y1059R) reduced the distance between the two juxtaposed α-helices from which their sidechains protrude (α1-helix: aa 983–998; α4-helix: aa 1048–1064, as named in *Lee et al., 2008*; *Figure 6C*), while, concomitantly, the loss of the salt bridges between E988 and K992 present in WT *Sc* Ire1-KR allows an increase in the distance between α1-helices from which both of these amino acid side chains protrude (*Figure 6D*). We used two metrics to quantify this change. First, we measured the distance between the centers of mass of the two α1-helices throughout the 20 ns simulation trajectories, which was increased by about 3 Å in *Sc* Ire1-KR(K992D,Y1059R) compared to WT *Sc* Ire1-KR. This measurement reflects the changes in protomer alignment caused by the salt bridge rewiring. Second, we measured the distance between the α carbons of R1039 involved in RNA substrate binding and H1061' involved in phosphodiester bond hydrolysis (*Korennykh et al., 2011*), which was decreased by about 5 Å in *Sc* Ire1-KR(K992D,Y1059R) compared to WT *Sc* Ire1-KR (*Figure 6E,F*). Therefore, the mutations that increase Ire1's RNase promiscuity are predicted to change both the RNase-RNase interface and the relative alignment of important elements in the catalytic site of the Ire1 dimer.

The kinase/RNase domain of Ire1 is homologous to the kinase homology (KH) and kinase extension nuclease (KEN) domains of the Ribonuclease L (RNase L), which mediates the antiviral and apoptotic effects of interferons in mammalian cells (*Chakrabarti et al., 2011*). Upon activation, RNase L forms homodimers (*Han et al., 2014*; *Huang et al., 2014*) and cleaves mRNAs with a sequence motif of UN|N, where N can be any ribonucleotide (*Han et al., 2014*). Since RNase L has promiscuous RNase activity, we wondered if its protomer alignment would resemble that of the *Sc* Ire1-KR (K992D,Y1059R). To test this notion, we compared the crystal structure of RNase L (PDB: 4OAV) with the structures of *Sc* Ire1-KR and *Sc* Ire1-KR(K992D,Y1059R). At the RNase-RNase dimer interface of RNase L, N601' and E671 form a pair of intermolecular hydrogen bonds between RNase L's α1- and α4-helix (*Figure 6—figure supplement 1A*; *Han et al., 2014*) (in RNase L the α1-helix spans aa W589-V599 and the α4-helix aa V659-H672, as defined in *Huang et al., 2014*). Thus, RNase L and *Sc* Ire1-KR(K992D,Y1059R) share a conserved arrangement by which their α1- and α4-helix are connected. By contrast, *Sc* Ire1-KR is different in that two α1-helices, one contributed by each protomer, are connected instead. In further support of this notion, we analyzed RNase L using the metrics defined in *Figure 6E* and found that the protomer alignment of RNase L closely resembles that of *Sc* Ire1-KR(K992D,Y1059R) (*Figure 6E*, *Figure 6—figure supplement 1A & B*). Thus, RNase L and *Sc* Ire1-KR(K992D,Y1059R), both of which have promiscuous RNase activity, share a similar protomer alignment.

## Discussion

From both an evolutionary and mechanistic angle, it has long been a puzzle how two modalities of Ire1 function arose and are structurally implemented. At one extreme lies Ire1 from *S. cerevisiae*, which is highly specific, precisely cleaving but a single mRNA in the cell (*HAC1* mRNA) at its two splice junctions to excise the intron and initiate mRNA splicing. At the other extreme lies Ire1 from *S. pombe*, which is highly promiscuous cleaving numerous mRNAs at recognition sites that share but a three-nucleotide consensus in a variably sized loop. Cleavage in this case initiates mRNA breakdown by RIDD. We previously showed by domain swapping experiments that Ire1's cytosolic kinase/RNase domains determine whether Ire1 works in the specific *S. cerevisiae*-like or relatively non-specific *S. pombe*-like modality. Using a growth assay based on heterologous expression of Ire1 kinase/RNase domains in bacteria that reports on Ire1's RNase promiscuity, we found that the *S. cerevisiae* Ire1's RNase specificity becomes promiscuous when only two amino acids, K992 and Y1059, are

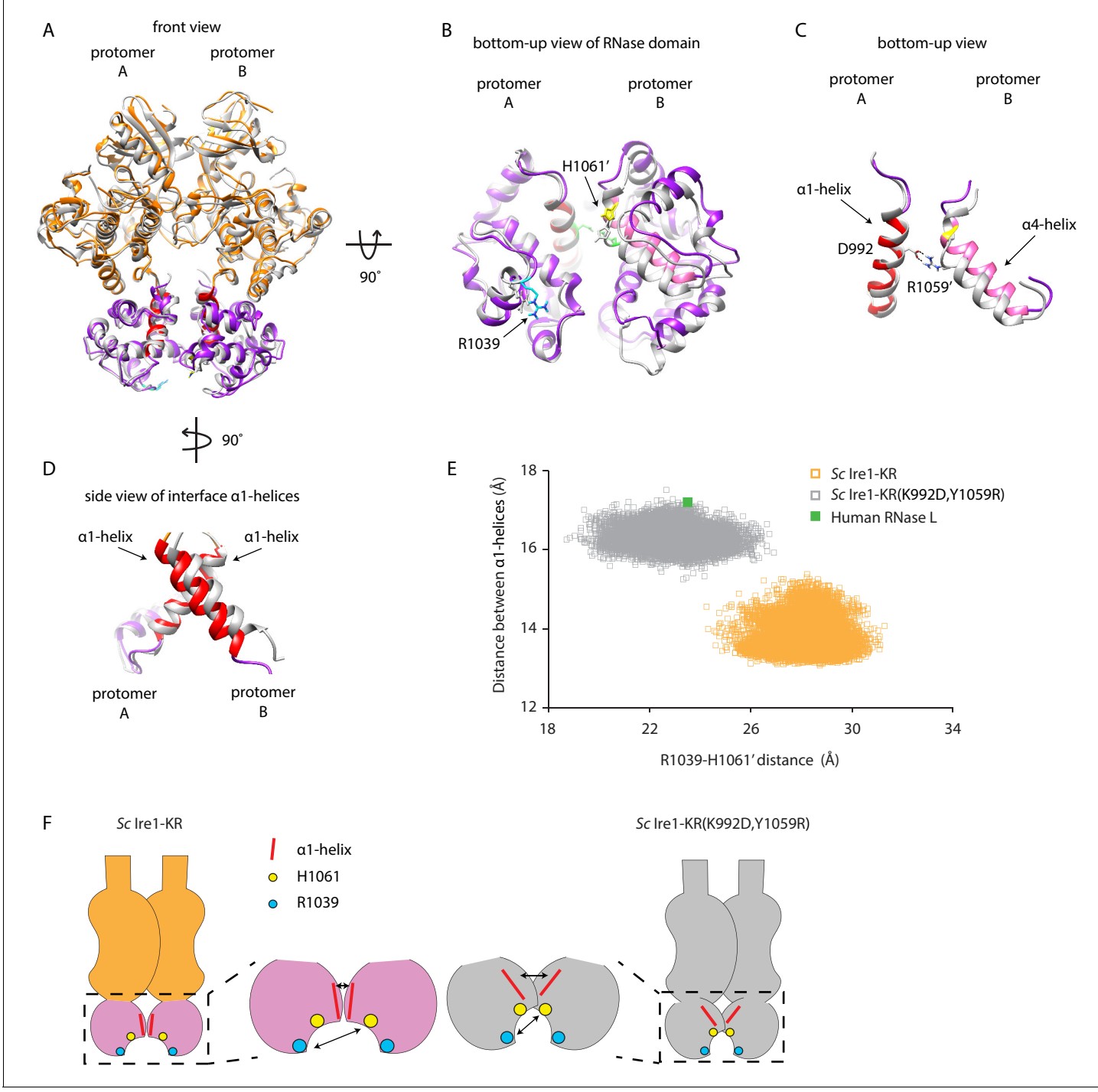

**Figure 6.** Interface mutations change the protomer alignment in Ire1 dimer. (**A**) Structure alignment of WT *Sc* Ire1-KR and *Sc* Ire1-KR(K992D,Y1059R). 20-ns MD simulations were performed on both WT *Sc* Ire1-KR and *Sc* Ire1-KR(K992D,Y1059R). The last simulation frame was used for structure alignment. The protomer A of the two dimers were aligned with minimal root mean square deviation. *Sc* Ire1-KR(K992D,Y1059R) is in gray. WT *Sc* Ire1-KR has its kinase domain in yellow and RNase domain in purple. The α1-helix at position 983–998, α4-helix at position 1048–1064, H1061, and R1039 are in red, pink, yellow, and cyan, respectively. (**B, C**) Bottom-up view of the aligned RNase domains (**B**), and the α1- and α4-helix (**C**). Side chains of D992 and R1059 are shown. Dashed lines are salt bridges. (**D**) Side view of α1-helices. Color coding are the same as in (**A**). Side chain labels on protomer B are marked with '. (**E**) Measuring Ire1 protomer alignment in the MD simulation. Y-axis is the distance between the centers of mass of the two α1-helices. X-axis is the distance between the α-carbons of R1039 on protomer A and H1061' on protomer B. Each dot represents a time point in the MD simulation. Measurements of WT *Sc* Ire1-KR are in yellow while measurements of *Sc* Ire1-KR(K992D,Y1059R) are in gray. The green dot is the measurement from the crystal structure of human RNase L (PDB: 4OAV). The R1039-H1061' distance on *Sc* Ire1 corresponds to the R651-H672' distance

*Figure 6 continued on next page*

*Figure 6 continued*

on RNase L. The α1-helix on RNase L is from W589 to V599. (F) Cartoon illustration showing the protomer alignment of WT *Sc* Ire1-KR and *Sc* Ire1-KR (K992D,Y1059R). Their RNase domains were zoomed in with double-arrow lines showing the distances being measured in (E).

The online version of this article includes the following figure supplement(s) for figure 6:

**Figure supplement 1.** Protomer alignment of RNase L.

replaced by aspartate and arginine respectively, which are the corresponding amino acids in *S. pombe* Ire1. While these replacements rendered *Sc* Ire1 more promiscuous, they did not entirely switch *Sc* Ire1's substrate RNA profile to that of *Sp* Ire1: *Sc* Ire1-KR(K992D,Y1059R) retained activity towards *HAC1* mRNA-derived stem-loops, which is inert to cleavage by *Sp* Ire1 (*Li et al., 2018*). The K992D and Y1059R mutations therefore rendered *Sc* Ire1 even more promiscuous than *Sp* Ire1.

MD simulations and biochemical assays revealed that the identified residues cause rewiring of two symmetry-related inter-molecular salt-bridges at Ire1's RNase-RNase interface within the back-to-back dimer. Sequence comparison of Ire1 orthologs showed that this rewiring is conserved and, where known, correlates with reported functional outputs (*Cheon et al., 2011*; *Hollien et al., 2009*; *Hollien and Weissman, 2006*; *Miyazaki and Kohno, 2014*; *Miyazaki et al., 2013*). By this criterion, the vast majority of species is predicted to have RIDD-enabled Ire1s as indicated by their *S. pombe*-like salt bridge pattern. Even though human Ire1α and Ire1β have different RNase specificity (*Imagawa et al., 2008*), both have a *S. pombe*-like interface pattern. This characteristic is consistent with both human Ire1 isoforms being able to perform RIDD (*Hollien et al., 2009*; *Iwawaki et al., 2001*). Of note, the ability of *Sc* Ire1-KR(K992D,Y1059R) to conduct both *HAC1*-specific and promiscuous cleavage resembles that of human Ire1α in its fully phosphorylated, oligomeric state, while the more restricted activity of *Sc* Ire1 resembles that of dimeric human Ire1α (*Le Thomas et al., 2021*).

Surprisingly, our work identified the RNase-RNase interface, rather than regions involved in substrate binding or catalysis, as a determinant for Ire1's RNase specificity. We show evidence that conserved salt bridges determine the relative protomer alignment. In the composite RNase active site of the back-to-back dimer, the relative distance of residues that contribute to cleavage from both protomers is changed (*Korennykh et al., 2011*). The small molecule, quercetin, which stabilizes *S. cerevisiae* Ire1's dimers/oligomers and increase its RNase activity (*Wiseman et al., 2010*), binds to the same site where the two mutations identified in this study are located. However, quercetin binding does not change Ire1 protomer alignment and hence is not expected to alter Ire1's RNase specificity (*Wiseman et al., 2010*). Nevertheless, modulation of the RNase selectivity by targeting the quercetin pocket is conceivable. In human Ire1, crystal structures showed that the two RNase domains in the dimer are further apart in the inactive state than in the active state (*Joshi et al., 2015*), and our data similarly indicate that breaking the salt bridges in *Sc* Ire1-KR(K992D,Y1059R) by either changing the aspartate or the arginine to alanine leads to profound reduction of activity. Related work demonstrates that the more promiscuous RIDD modality of human Ire1α requires phosphorylation-driven oligomerization, which can be prevented by an oligomer-disrupting mutation at the RNase-RNase interface within the back-to-back dimers (*Le Thomas et al., 2021*). Thus together, the data presented here demonstrate that Ire1 RNase domain's dimer interface is a dynamic site through which both activity and substrate specificity can be regulated.

## Materials and methods

### Recombinant protein expression and purification

All the plasmids used in this study are listed in *Table 1*. The cytoplasmic portion of *Sc* or *Sp* Ire1 containing its kinase and RNase domains (Ire1-KR) was expressed and purified from BL21-CodonPlus (DE3)-RIPL *Escherichia coli*. We used an expression vector which fuses a PreScission Protease cleavage site between the Ire1-KR and glutathione *S*-transferase (GST) domains of the recombinant polypeptide and was regulated by a T7 promoter. The expression cassette was transformed into *E. coli* cells. The WT *Sc* Ire1-KR was expressed as described previously (*Korennykh et al., 2009*). For *E. coli* cells transformed with plasmids containing the *Sp* Ire1-KR or *Sc* Ire1-KR mutant, all colonies on the transformation plate were collected 16 hr after transformation and mixed with 50 mL of LB medium. After 3 hr incubation at 37°C, the sample was diluted to 12 L of LB medium and further incubated at

**Table 1.** Plasmids used in this study.
In all of the plasmids, a GST and an HRV 3C protease site are N-terminally fused to Ire1-KR.

| Plasmid number | Description | Source |
| --- | --- | --- |
| pPW1477 | *Sc* Ire1-KR on pGEX6P-2 | *Korennykh et al., 2009* |
| pPW3205 | *Sp* Ire1-KR on pGEX6P-2 | *Li et al., 2018* |
| pPW3244 | *Sc* Ire1-KR-mut17 on pGEX6P-2 | This study |
| pPW3262 | *Sc* Ire1-KR(K992D,Y1059R) on pGEX6P-2 | This study |
| pPW3263 | *Sc* Ire1-KR(K992D,H1044D,Y1059R) on pGEX6P-2 | This study |
| pPW3256 | revertant 1 (K992) on pGEX6P-2 | This study |
| pPW3245 | revertant 2 (N1001) on pGEX6P-2 | This study |
| pPW3246 | revertant 3 (M1010) on pGEX6P-2 | This study |
| pPW3247 | revertant 4 (T1032) on pGEX6P-2 | This study |
| pPW3248 | revertant 5 (F1033) on pGEX6P-2 | This study |
| pPW3257 | revertant 6 (E1038) on pGEX6P-2 | This study |
| pPW3258 | revertant 7 (R1039) on pGEX6P-2 | This study |
| pPW3259 | revertant 8 (H1044) on pGEX6P-2 | This study |
| pPW3260 | revertant 9 (S1045) on pGEX6P-2 | This study |
| pPW3249 | revertant 10 (M1049) on pGEX6P-2 | This study |
| pPW3250 | revertant 11 (Y1059) on pGEX6P-2 | This study |
| pPW3261 | revertant 12 (F1062) on pGEX6P-2 | This study |
| pPW3251 | revertant 13 (M1063) on pGEX6P-2 | This study |
| pPW3252 | revertant 14 (I1069) on pGEX6P-2 | This study |
| pPW3253 | revertant 15 (A1070) on pGEX6P-2 | This study |
| pPW3254 | revertant 16 (E1071) on pGEX6P-2 | This study |
| pPW3255 | revertant 17 (L1109) on pGEX6P-2 | This study |
| pPW3441 | *Sc* Ire1-KR(K992D,Y1059A) on pGEX6P-2 | This study |
| pPW3442 | *Sc* Ire1-KR(K992A,Y1059R) on pGEX6P-2 | This study |
| pPW3443 | *Sc* Ire1-KR(K992R,Y1059D) on pGEX6P-2 | This study |
| pPW3275 | *Sc* Ire1-KR(H1018A) on pGEX6P-2 | This study |

37°C until optical density reached 1. The incubation temperature was reduced to 25°C and protein expression was induced by adding IPTG to a final concentration of 0.5 mM. After 4 hr of growth at 25°C, the cells were pelleted by centrifugation.

Cells were resuspended in GST binding buffer (50 mM Tris-HCl pH 7.5, 500 mM NaCl, 2 mM Mg(OAc)$_2$, 2 mM DTT, 10% glycerol) and homogenized using high-pressure homogenizer (EmulsiFlex). The cell lysate was applied to a GST-affinity column and eluted with GST elution buffer (50 mM Tris-HCl pH 7.5, 200 mM NaCl, 2 mM Mg(OAc)$_2$, 2 mM DTT, 10% glycerol, 10 mM reduced glutathione). The column elution was treated with GST-tagged HRV 3C protease (PreScission Protease, GE Health) to cleave off the GST tag. At the same time, the sample was dialyzed to remove glutathione in the elution buffer. After 12 hr dialysis, the sample was further purified through negative chromatography by passing through a GST-affinity column (GSTrap FF Columns, GE Healthcare Life Sciences) to remove free GST, residual GST-fused Ire1 KR, and GST-tagged protease, and a Q FF anion exchange column (GE Healthcare Life Sciences) to remove contaminating nucleic acids. The flow-through containing untagged Ire1 KR was further purified by applying it to a Superdex 200 16/60 gel filtration column (GE healthcare) and then concentrated to 20–40 µM in storage buffer (50 mM Tris-HCl pH 7.5, 500 mM NaCl, 2 mM Mg(OAc)$_2$, 2 mM TCEP, 10% glycerol) and flash-frozen in liquid nitrogen. The final purity, as well as purity at intermediate steps, was assessed by SDS-PAGE using Coomassie blue staining.

## In vitro RNA cleavage assay

Short RNA oligos derived from the Ire1 cleavage sites on *Sp BIP1* mRNA, *SPAC4G9.15* mRNA, *PLB1* mRNA, and the *Sc HAC1* mRNA 3' splice site were purchased from Dharmacon, Inc The sequence of stem-loop RNA substrates ordered were the following: *Sp BIP1* cleavage site: 5'CGCGAGAUAAC UGGUGCUUUGUUAUCUCGCG, *Sp SPAC4G9.15* cleavage site: 5'CCACCACCGAGUAUGCUAC UCGGUGGUGG, *Sp PLB1* cleavage site: 5'ACGGCCUUUGUUGCAAAAGGGUCGU (25 bp), and *Sc HAC1* 3' splice site: 5'GCGCGGACUGUCCGAAGCGCAGUCCGCGC. RNA oligos were gel extracted, acetone precipitated, and resuspended in RNase-free water. The oligos were 5'-end radiolabeled with gamma-[$^{32}$P]-ATP (Perkin Elmer) using T4 polynuclotide kinase (NEB) and cleaned using ssDNA/RNA Clean and Concentrator kit (Zymo Research D7010).

To fold the RNA, we heated the RNA oligos to 90℃ for 3 min and slowly cooled them down at a rate of 1℃ per minute until the temperature reached 10℃. In the Ire1 cleavage assays, the reaction samples contained 12.5 µM of Ire1-KR. The cleavage reaction was performed as described previously (*Li et al., 2018*) by incubating at 30℃ in reaction buffer (50 mM Tris/HCl pH 7.5, 200 mM NaCl, 2 mM Mg(OAc)$_2$, 2 mM TCEP, 10% glycerol). For reactions in *Figure 1B & C*, 200 µM of 4µ8C (Sigma-Aldrich) was added. At each time point, an aliquot was transferred to 1.2x STOP buffer (10 M urea, 0.1% SDS, 1 mM EDTA, trace amounts bromophenol blue). RNAs were separated using denaturing 15% Novex TBE-Urea Gels (ThermoFisher) and transferred to Amersham Hybond-N + membranes. Radioactive RNA membranes were imaged with a Phosphorimager (Typhoon FLA 9500, GE Health). The band intensities were quantified using ImageJ. The cleaved portion was calculated as the cleaved band intensity divided by the sum of the cleaved band and uncleaved band intensities. The $k_{obs}$ values were obtained by fitting the data to first-order 'one-phase' decay equations using Prism (GraphPad).

## In vitro RNA cleavage assay of *HAC1*- and *BIP1*-derived RNA mutants

In vitro transcription of the mutant RNA stem-loops derived from the *HAC1* 3' splice site and *BIP1* cleavage site were carried out as follows. Singe-stranded DNA oligonucleotides were used as templates to which the 18mer 5'TAATACGACTCACTATAG 'T7 promoter oligonucleotide' was annealed to create a double-stranded T7 RNA polymerase promoter. The templates contain the indicated single point mutations from *Figure 4* on the following parent oligonucleotides: HAC1-27 (encoding wild-type *HAC1* 3' stem-loop RNA with T7 promoter): 5'GCGCGGACTGCGTTCGGACAGTCCGCC TATAGTGAGTCGTATTA, and BIP1-32 (encoding wild-type *BIP1* stem-loop RNA with T7 promoter): 5'CGCGAGATAACAAAGCACCAGTTATCTCGCGCTATAGTGAGTCGTATTA.

A solution containing 5 nM T7 promoter oligonucleotide and 0.75 nM template oligonucleotide was heated to 100℃ for 3 min and immediately placed on ice. 20 µL transcription reactions containing 5 µL of the template solution, 1 mM each of ATP, CTP, GTP, and UTP, 1x reaction buffer, and 2 µL T7 RNA Polymerase mix (HiScribe T7 High Yield RNA Synthesis Kit, NEB) were incubated at 37℃ for 3 hr. RNA oligos were gel extracted in 300 µL RNase-free water. These RNA substrates are not radio labeled. The RNAs are folded, cleaved by Ire1-KR and separated by TBE-Urea gels in the same way as the radio labeled RNAs. The TBE-Urea gels were stained with SYBR Gold (ThermoFisher) and imaged on the Typhoon with excitation at 488 nm. The emission was collected using a band pass filter at 550 nm. Image analysis is the same as radio labeled RNAs.

## Bacterial growth assays

Expression vectors containing the *Sc* Ire1-KR, *Sp* Ire1-KR, or a mutant form of these proteins regulated by a T7 promoter were transformed into BL21-CodonPlus (DE3)-RIPL *E. coli* cells. Freshly transformed *E. coli* cells were cultured overnight (~20 hr) and then diluted to an OD$_{600}$ of 0.02. The cultures were incubated at 37℃ and their OD$_{600}$ was obtained every 15 min by the Tecan Spark Multimode Microplate Reader (Life Sciences) (60 cycle kinetic loop, continuous shaking, double orbital 2.5 mm, 108 rpm) for 5 hr. For cultures containing the Ire1 RNase inhibitor 4µ8C, 1 µM of 4µ8C (Sigma-Aldrich) was added into both the overnight culture and the diluted culture. It is important to use freshly transformed (transformed within 72 hr) *E. coli* cells as the toxicity of *Sp* Ire1-KR and mutants of *Sc* Ire1-KR accumulates over time.

## Crosslinking gel

Each Ire1 construct was buffer-exchanged three times with Zeba spin desalting columns (Thermo-Fisher Scientific 89882) into a buffer containing 50 mM HEPES, 200 mM NaCl, 2 mM Mg(OAc)$_2$, 2 mM TCEP, and 10% Glycerol. 8 µL crosslinking reactions containing 12.5 µM Ire1, 2 mM ADP, and 1 mM BS3 crosslinker (ThermoFisher Scientific 21580) were carried out on ice for two hours and quenched by adding concentrated Tris-HCl to a final concentration of 60 mM. The entire reaction was separated on an SDS-PAGE gel, stained with SYPRO Ruby (ThermoFisher Scientific S21900) or coomassie blue overnight and scanned with the Typhoon FLA 9500 or Gel Documentation system (Bio-Rad) respectively.

## Analytical ultracentrifugation

Experiments were performed in a Beckman Coulter Optima XL-A analytical ultracentrifuge equipped with UV-visible absorbance detection system using a 4-hole An-60 Ti analytical rotor. Multi-speed sedimentation equilibrium experiments were carried out at 20°C and 7,000, 10,000, and 14,000 rpm until equilibrium was reached for 110 µL samples of concentrations of 10 µM, 5 µM, and 2.5 µM protein. Samples were dialyzed overnight into analysis buffer (50 mM Tris/HCl, pH 7.5, 200 mM NaCl, 2 mM Mg(OAc)$_2$, 2 mM TCEP) to remove glycerol. Analysis buffer without protein was used as reference. Measurements were made at 280 nm in the absorbance optics mode. Raw data were trimmed using WinReEdit (Jeff W. Lary, University of Connecticut) and globally fitted to a self-association equilibrium model using WinNonlin (*Johnson et al., 1981*) using all concentrations and speeds for each protein sample.

## Molecular dynamics simulation

All simulations were performed using the Amber suite (*Case et al., 2014*). Initial structure and topology files were prepared in LEaP (*Sastry et al., 2013*) using the Amber ff12SB force field, the general Amber force field (gaff) and the phosaa10 parameters for phosphorylated amino acids. The structural model included an inhibitor bound to the kinase domain that was kept in place for the simulations. Inhibitor parametrization was performed using AnterChamber (*Wang et al., 2006*; *Wang et al., 2004*). We solvated the system with TIP3P water molecules in a periodic cubic box such that the closest distance between the periodic boundary and the closest atom in the protein was 10 Å. We added counterions to neutralize the box.

We minimized the energy of the system, first using harmonic restraints on the protein backbone (10 kcal mol$^{-1}$ Å$^{-2}$) then in an unrestrained minimization, for 500 steps of steepest descent and 500 steps of conjugate gradient, each at constant volume with a non-bonded cutoff distance of 9 Å.

We performed a three-step equilibration: 1. heating the system to 300 K at constant volume with harmonic restraints on the protein backbone (10 kcal mol$^{-1}$ Å$^{-2}$) at constant volume using the SHAKE algorithm (*Ciccotti and Ryckaert, 1986*) constraining bonds involving hydrogens and the Andersen thermostat for 20 ps; 2. constant pressure of 1 bar with lower restraints on the protein backbone (one kcal mol$^{-1}$ Å$^{-2}$) for 20 ps with consistent parameters; 3. the restraints on the protein backbone were released and the system was equilibrated at constant pressure for one ns.

We seeded the production runs with random new velocities at constant pressure of 1 bar and a non-bonded cutoff distance of 9 Å and ran the molecular dynamics simulations for 20 ns with a two fs time step. Coordinates and energy were saved every picosecond (500 steps). We assessed the convergence of the simulations by examining backbone root-mean-square deviation (RMSD) plots with particular focus on the RNase domain.

Visual inspection of molecular models was performed using ChimeraX (*Goddard et al., 2018*), a virtual reality implementation of the traditional molecular visualization program. Molecular graphics and analyses performed with UCSF Chimera (*Pettersen et al., 2004*), developed by the Resource for Biocomputing, Visualization, and Informatics at the University of California, San Francisco, with support from NIH P41-GM103311.

## Ire1 deep alignment

Ire1 orthologs were identified by searching the Pfam Ribonuc_2-5A hidden Markov model (HMM) with HMMSEARCH from HMMER 3.1b2 (*Eddy, 2011*) against all proteins from RefSeq fungal genomes downloaded from GenBank 1/17/2017 plus six plant genomes (*C. elegans*, *D.

*melanogaster*, *D. rerio*, *M. musculus*, *M. mulatta*, and *H. sapiens*) and six animal genomes (*A. thaliana*, *C. reinhardtii*, *P. patens*, *O. sativa*, *S. lycopersicum*, and *P. glauca*) downloaded from GenBank 7/19/2019. This gave predominantly hits to Ire1 orthologs. Outlier hits to RNaseL and difficult to place microsporidian RNases were removed. A guide multiple alignment was generated by PROB-CONS 1.12 alignment (*Do et al., 2005*) of the Ire1 hits from plants and animals plus 14 fungal species (*H. capsulatum*, *A. nidulans*, *N. crassa*, *F. graminearum*, *Y. lipolytica*, *S. cerevisiae*, *C. albicans*, *K. lactis*, *S. pombe*, *P. carnii*, *C. neoformans*, *U. maydis*, *P. graminis*, and *M. verticillata*). HMMBUILD was used to generate an HMM from the conserved regions of the PROBCONS alignment, spanning the kinase and RNase domains. HMMALIGN was then used to realign the full set of Ire1 hits to the new HMM. A maximum likelihood phylogeny was estimated from the aligned positions with FAST-TREE 2.1.8 (*Price et al., 2010*). The resulting tree was rendered with an in-house PYTHON script.

## Acknowledgements

We thank Avi Ashkenazi, Adrien Le Thomas, Wallace Marshall, Jonathan Weissman, Marc Shuman and members of the Walter lab for their insightful discussions. In particular we thank Alexei Korennykh (self-identified as Reviewer #2) for suggesting that we extend the structural prediction to include a comparison with RNase L (included in *Figure 6*). This work was supported by UCSF-Zaffaroni Fellowship (WL), the Human Frontier Science Program (JP), NIH/NIAID 2R37 AI066224 (AS), NIH R01 GM032384 (PW), 1R35 GM118119-01 (DW), Howard Hughes Medical Institute EXROP (CR). JP is an Emmy Noether Fellow of the Deutsche Forschungsgemeinschaft. PW and DM are Investigators of the Howard Hughes Medical Institute.

## Additional information

### Funding

| Funder | Grant reference number | Author |
| --- | --- | --- |
| Howard Hughes Medical Institute | Investigator Grant | R Dyche Mullins<br>Peter Walter |
| Deutsche Forschungsgemeinschaft | Emmy Noether fellow | Jirka Peschek |
| University of California, San Francisco | UCSF-Zaffaroni Fellowship | Weihan Li |
| Human Frontier Science Program | | Jirka Peschek |
| National Institute of Allergy and Infectious Diseases | 2R37 AI066224 | Anita Sil |
| National Institute of General Medical Sciences | R01 GM032384 | Peter Walter |
| National Institute of General Medical Sciences | 1R35 GM118119-01 | R Dyche Mullins |

The funders had no role in study design, data collection and interpretation, or the decision to submit the work for publication.

### Author contributions

Weihan Li, Conceptualization, Data curation, Formal analysis, Validation, Investigation, Methodology, Writing - original draft, Writing - review and editing; Kelly Crotty, Data curation, Formal analysis, Validation, Investigation, Methodology, Writing - original draft, Writing - review and editing; Diego Garrido Ruiz, Data curation, Validation, Investigation, Methodology, Writing - review and editing; Mark Voorhies, Data curation, Investigation, Methodology, Writing - review and editing; Carlos Rivera, Data curation, Investigation, Writing - review and editing; Anita Sil, Supervision, Funding acquisition, Writing - review and editing; R Dyche Mullins, Matthew P Jacobson, Funding acquisition, Methodology, Writing - review and editing; Jirka Peschek, Formal analysis, Supervision, Investigation, Methodology, Writing - original draft, Writing - review and editing; Peter Walter,

Conceptualization, Supervision, Funding acquisition, Investigation, Writing - original draft, Writing - review and editing

### Author ORCIDs
Weihan Li https://orcid.org/0000-0003-4718-1884
Diego Garrido Ruiz http://orcid.org/0000-0002-2441-385X
Mark Voorhies http://orcid.org/0000-0001-8815-7384
Jirka Peschek https://orcid.org/0000-0001-8158-9301
Peter Walter https://orcid.org/0000-0002-6849-708X

### Decision letter and Author response
Decision letter https://doi.org/10.7554/eLife.67425.sa1
Author response https://doi.org/10.7554/eLife.67425.sa2

## Additional files

### Supplementary files
• Transparent reporting form

### Data availability
All data generated or analysed during this study are included in the manuscript and supporting files.

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
