## [Decision Letter]

**Acceptance summary:-**

This work answers the long-standing question concerning the specificity of the receptor Ire1, which is responsible for resolving protein misfolding stress in eukaryotes and antibody production in mammals. The work should be impactful for the fields of protein folding stress and antiviral dsRNA sensing mediated by the Ire1 paralog RNase L.

**Decision letter after peer review:**

Thank you for submitting your article "Protomer alignment modulates specificity of RNA substrate recognition by Ire1" for consideration by *eLife*. Your article has been reviewed by 2 peer reviewers, and the evaluation has been overseen by a Reviewing Editor and Vivek Malhotra as the Senior Editor. The following individual involved in review of your submission has agreed to reveal their identity: Alexei V Korennykh (Reviewer #2).

Essential revisions:

This is a very strong paper. Please pay attention to the reviewers' suggestions enumerated below. Please consider also changing the subtitle "Promiscuous RNase activity of *S. pombe* Ire1 is toxic to bacterial cells" and including in the discussion the recent data on the structure of RNAse L and its relevance to this paper.

*Reviewer #1:*

The manuscript by Li et al. presents development of assay to study substrate specificity of UPR-related Ire1 RNAse as well as structure-sequence functional comparison of Ire1s from two yeasts. The manuscript does not have weaknesses and clarifies additional questions about substrate specificity in the field.

While I understand where the authors come from with first subtitle "Promiscuous RNase activity of *S. pombe* Ire1 is toxic to bacterial cells" I am not quite sure that this is the correct explanation of the data shown in the figure 1.

It could be that activity of *S. pombe* Ire1 is not really promiscuous but specific towards ie. ribosomal hairpins or essential mRNAs in *E. coli* cells. This would also be toxic to bacterial cells. I recommend that they change maybe a subtitle to just "RNase activity of *S. pombe* Ire1 is toxic to bacterial cells". This is supported with figure 1.

*Reviewer #2:*

This work focuses on a single question: what determines RNA recognition by the UPR receptor Ire1. This question is fundamentally important due to the key role of Ire1 in the UPR pathway and due to surprising difficulty of understanding specificity switching in related enzymes, generally. Similar studies in the past were successful only in a handful of examples.

The authors for the first time re-engineer specificity of Ire1 and convert "splicing" Ire1 into "RNA decay" Ire1. This work suggests that RNase/RNase interface contacts control the RNA preference, providing the molecular basis to guide our thinking about Ire1 activity and potentially elucidate evolution of the antiviral receptor RNase L from Ire1.

This is a logically designed and well-presented study.

1. The specificity of Sp Ire1 for UG|C resembles the specificity of the related protein RNase L, which cleaves UN|N sequences, including UG|C.

2. A few crystal structures of RNase L have been solved. Can the authors comment on these structures? For example, when placed on Figure 6E, does RNase L fall in Sc-like or Sp-like cloud?

---

## [Author Response]

Reviewer #1:The manuscript by Li et al. presents development of assay to study substrate specificity of UPR-related Ire1 RNAse as well as structure-sequence functional comparison of Ire1s from two yeasts. The manuscript does not have weaknesses and clarifies additional questions about substrate specificity in the field.While I understand where the authors come from with first subtitle "Promiscuous RNase activity of *S. pombe* Ire1 is toxic to bacterial cells" I am not quite sure that this is the correct explanation of the data shown in the figure 1.It could be that activity of S. pombe Ire1 is not really promiscuous but specific towards ie. ribosomal hairpins or essential mRNAs in *E. coli* cells. This would also be toxic to bacterial cells. I recommend that they change maybe a subtitle to just "RNase activity of *S. pombe* Ire1 is toxic to bacterial cells". This is supported with figure 1.

We have changed the subtitle as suggested.

Reviewer #2:This work focuses on a single question: what determines RNA recognition by the UPR receptor Ire1. This question is fundamentally important due to the key role of Ire1 in the UPR pathway and due to surprising difficulty of understanding specificity switching in related enzymes, generally. Similar studies in the past were successful only in a handful of examples.The authors for the first time re-engineer specificity of Ire1 and convert "splicing" Ire1 into "RNA decay" Ire1. This work suggests that RNase/RNase interface contacts control the RNA preference, providing the molecular basis to guide our thinking about Ire1 activity and potentially elucidate evolution of the antiviral receptor RNase L from Ire1.This is a logically designed and well-presented study.1. The specificity of Sp Ire1 for UG|C resembles the specificity of the related protein RNase L, which cleaves UN|N sequences, including UG|C.

We thank the reviewer for this suggestion. By comparing the crystal structure of RNase L (PDB: 4OAV) with our MD-derived model of *Sc* Ire1-KR(K992D,Y1059R), we found that the dimer interface of RNase L and *Sc* Ire1-KR(K992D,Y1059R) are similar as their α1- and α4-helix form intermolecular contacts. By contrast, wild-type *Sc* Ire1-KR is different as solely its α1-helices interact instead. Furthermore, using the metrices defined in Figure 6E, we found that the protomer alignment of RNase L is similar to that of the *Sc* Ire1-KR(K992D,Y1059R). Thus, in line with RNase L having promiscuous RNase activity, the protomer alignment of RNase L resembles that of the more promiscuous *Sc* Ire1-KR(K992D,Y1059R) variant. New data are in Figure 6E and Figure 6—figure supplement 1.

2. A few crystal structures of RNase L have been solved. Can the authors comment on these structures? For example, when placed on Figure 6E, does RNase L fall in Sc-like or Sp-like cloud?

We have now included the experimental replicates in Figure 1A.